# Microbiome Modulation in Pediatric Leukemia: Impact on Graft-Versus-Host Disease and Treatment Outcomes: A Narrative Review

**DOI:** 10.3390/children12020166

**Published:** 2025-01-29

**Authors:** Samuel Bogdan Todor, Cristian Ichim

**Affiliations:** Faculty of Medicine, University Lucian Blaga of Sibiu, 550024 Sibiu, Romania; cristian.ichim@ulbsibiu.ro

**Keywords:** childhood leukemia, gut microbiome, fecal microbiota transplant, hematopoietic stem-cell transplant, graft-versus-host disease

## Abstract

The gut microbiome significantly influences the outcomes of pediatric leukemia, particularly in patients undergoing hematopoietic stem cell transplantation (HSCT). Dysbiosis, caused by chemotherapy, antibiotics, and immune system changes, contributes to complications such as graft-versus-host disease (GVHD), gastrointestinal issues, and infections. Various microbiome-related interventions, including prebiotics, probiotics, postbiotics, and fecal microbiota transplantation (FMT), have shown potential in mitigating these complications. Specific microbial signatures have been linked to GVHD risk, and interventions like inulin, Lactobacillus, and SCFAs (short-chain fatty acids), particularly butyrate, may help modulate the immune system and improve outcomes. FMT, while showing promising results in restoring microbial balance and alleviating GVHD, still requires careful monitoring due to potential risks in immunocompromised patients. Despite positive findings, more research is needed to optimize microbiome-based therapies and ensure their safety and efficacy in pediatric leukemia care.

## 1. Introduction

Bacterial infections represent a significant challenge in pediatric oncology, particularly among children and adolescents undergoing induction chemotherapy for acute lymphoblastic leukemia (ALL) or acute myeloid leukemia (AML), as well as those receiving HSCT. Induction chemotherapy poses the highest risk for infection-related morbidity and mortality, accounting for nearly half of all infection-related deaths during leukemia treatment. Similarly, children undergoing HSCT are at an elevated risk due to prolonged periods of severe neutropenia (absolute neutrophil count <500/µL), chemotherapy-induced mucosal barrier injury, and the routine use of central venous catheters. These factors necessitate the prompt initiation of empiric antibiotic therapy at the first indication of fever, irrespective of clinical presentation or microbiological confirmation, underscoring the critical role of infection management in these vulnerable populations [1,2]. As an unwanted side effect, extended use of broad-spectrum antibiotics in children undergoing HSCT has been tied to the emergence of antibiotic-resistant bacteria [3]. Factors like prolonged neutropenia, breakthrough bacteremia, and prolonged hospitalization period predict antibiotic resistance [4]. Gram-negative bacteria show high levels of resistance to multiple antibiotics, including carbapenems, while methicillin-resistant S. aureus and vancomycin-resistant E. faecium are also prevalent in these individuals. Prior antibiotic use and colonization with resistant pathogens are key risk factors for bloodstream infections [5]. Another consequence of antibiotic use in children receiving HSCT is *Clostridioides difficile* infection. The use of proton pump inhibitors (PPIs) and third/fourth generation cephalosporins are the main predictors of this event [6]. The early use of antibiotics has also been associated with transplant-related mortality in allogeneic stem cell transplant (AHST) due to microbiome disruption. C-Commensal species from the *Clostridiales* group are involved [7]. Pretransplant stool in children undergoing HSCT showed significantly reduced microbial diversity compared to healthy children. The HSCT group had notably lower levels of *Bacteroides*, *Ruminococcaceae*, and butyrate-producing genes, while opportunistic pathogens from gammaproteobacterial species were more abundant [8].

Allogeneic HSCT (Allo-HSCT), primarily indicated as a curative treatment for hematologic malignancies, is notably associated with various complications, including infections, disease relapse, and GvHD [9]. Furthermore, changes in gut microbiota composition have also been linked to HSCT. Key contributors to these microbiota alterations include mucositis, recurrent Clostridioides difficile infections (RCDIs), GvHD, antibiotic administration, irradiation, and conditioning chemotherapy, all of which may arise following the allo-HSCT procedure [10]. Reduced gut microbiota diversity has been associated with poorer clinical outcomes and higher mortality following allo-HSCT, with an overgrowth of opportunistic bacteria such as *Enterococcus* and *Proteobacteria*, linked to increased infection risk and mortality [9,11]. Additionally, lower levels of SCFAs, particularly propionate and butyrate, were associated with greater microbiota imbalance, antibiotic use, and the development of GvHD [12].

The aim of this narrative review is to investigate the impact of gut microbiota diversity and alterations on the outcomes of HSCT and chemotherapy in pediatric patients with leukemia, with a particular emphasis on the effects of antibiotic use. The review examines the relationship between microbiota disruptions and clinical outcomes, including infection risk, GvHD, treatment-related mortality, and overall success or failure of HSCT and chemotherapy. Studies were selected from PubMed and ScienceDirect using the following key terms: “hematopoietic stem cell transplantation”, “chemotherapy”, “pediatric leukemia”, “microbiota diversity”, “antibiotic use”, “treatment outcomes”, “bacterial infections”, “antibiotic resistance”, “graft-versus-host disease”, “*Clostridioides difficile*”, “neutropenia”, and “bloodstream infections”. Only peer-reviewed articles were included that addressed the influence of microbiota diversity, the role of SCFAs, and the consequences of microbial alterations on treatment outcomes in pediatric leukemia.

## 2. Gut Microbiome Dynamics During Childhood

Maternal diet during pregnancy, including vegetable intake and omega-3 fatty acid supplementation, showed no significant impact on the infant gut microbiome in observational and randomized trials [13]. However, omega-3 long-chain polyunsaturated fatty acids (LCPUFAs) in fish oil may have potential anti-inflammatory effects, reducing oxidative stress and pathogenic microbes in the gut [14]. Mode of delivery influences early intestinal colonization in full-term infants, with vaginal delivery promoting beneficial microbes like *Lactobacillus* and *Bacteroides*, while cesarean delivery favors skin- and hospital-associated microbes such as *Klebsiella* and *Enterococcus*, potentially leading to delayed gut microbiota stability and higher respiratory infection risk in the first year of life. However, some studies suggest that these differences in gut flora diminish by six weeks of age [15,16,17]. A study found no association between ALL and cesarean delivery, regardless of whether it occurred during or before labor [18]. However, a recent meta-analysis indicated an increased risk of ALL in children delivered via elective cesarean section [19]. Regarding breastfeeding, research has shown that breastfeeding for six months or longer is associated with a reduced risk of ALL [20]. Infant diet shapes microbiome development, with breastfed infants harboring *Lactobacillus* and *Bifidobacterium*, while formula-fed infants have microbes like *Roseburia* and *Clostridium*, associated with inflammation. Cessation of breastfeeding accelerates microbiome maturation toward an adult-like composition, favoring microbes that degrade dietary fibers and produce SCFAs [15,21]. Microbiome development in preterm infants, primarily driven by gestational age, is dominated by Enterobacter, Staphylococcus, and Enterococcus, contrasting with the Bacteroides and Bifidobacterium-rich microbiome of full-term infants [22]. Gut microbiome (GM) development in children varies with age and dietary habits, showing compositional stability similar to adults by preschool and school age [23]. Studies reveal associations between dietary patterns and microbial composition, with certain foods influencing beneficial SCFA-producing bacteria like Bifidobacterium, but research on microbial development during puberty and in non-Western populations remains limited (Figure 1) [24].

Early antibiotic exposure alters gut microbiota, influencing body mass by increasing or decreasing weight, with effects depending on maternal weight; antibiotics increase overweight risk in infants of normal-weight mothers but decrease it in those of overweight mothers. Overweight children show higher *Staphylococcus aureus* levels, while normal-weight children have more *Bifidobacteria* in their gut microbiota (Figure 1) [25,26].

## 3. Microbiome Implications in Acute Leukemia in Children

The development of B-cell precursor acute lymphoblastic leukemia (pB-ALL) involves genetic susceptibility, chromosomal aberrations, and environmentally driven mutations, with infections acting as progression-enhancing stimuli [27]. Studies in predisposed mice exposed to pathogens like murine norovirus and Helicobacter species showed increased leukemia incidence compared to pathogen-free conditions. Notably, predisposed mice did not develop significantly more cancer than wild-type mice without infective stimuli, highlighting the critical role of infections in leukemia progression [28]. Recent research suggests that the GM plays a role in the development of acute leukemia (AL), with early microbial exposures causing lasting changes that can influence immune responses and disease progression [12]. In a mouse model of B-cell precursor ALL (pB-ALL) with genetic mutations Pax5+/− and ETV6-RUNX1, the GM was altered according to these mutations, and mice raised in pathogen-free conditions did not develop ALL. These findings suggest that an intact GM may protect genetically predisposed individuals from developing ALL, and alterations in the GM may trigger disease onset; however, these are results from experimental studies [29] (Figure 2).

Pediatric patients with ALL exhibit distinct gut microbiome features, including a higher prevalence of *Bacteroides clarus*, *Roseburia faecis*, *Edwardsiella tarda*, and *Fusobacterium naviforme*, compared to healthy controls [30]. These patients also show lower alpha diversity and a higher relative abundance of Bacteroidetes, with reduced Firmicutes and Actinobacteria, though prior antibiotic use may influence these findings [31]. Reduced alpha diversity and enrichment of *Anaerostipes*, *Coprococcus*, *Roseburia*, and *Ruminococcus* have been observed, regardless of antibiotic exposure, alongside a higher prevalence of *Megamonas* and a lower prevalence of *Blautia* [32,33]. Additionally, imbalances in the oral microbiome suggest a potential role for microbiomes from other anatomical sites in disease progression [34]. Some reports consistently found lower gut microbiome α-diversity, including Shannon, Simpson, and Chao1 indices, in children with ALL compared to controls, though matching by age and antibiotic exposure varied. Differences in β-diversity of the gut microbiome between healthy children and children with ALL at the time of diagnosis reported a statistically significant Bray–Curtis dissimilarity between the two groups [35] (Figure 2).

Studies reveal that children with ALL at diagnosis have reduced α-diversity and altered β-diversity in the gut microbiome, with a notable decrease in Firmicutes, a phylum critical for gut microbiome maturation during early development. Additionally, they show reduced relative abundance of genera associated with older developmental trajectories, which typically expand after weaning (less than 1%). These findings suggest a significant disruption in normal gut microbiome maturation compared to healthy children [15].

The consistent reduction in specific bacterial taxa across diverse studies suggests a pervasive lag in gut microbiome maturation in children with ALL, likely originating from adverse exposures during the first year of life [35] (Figure 2).

Although less common in children, AML shows similar microbiome changes as seen in ALL [36]. In adult AML patients, *Actinomyces*, *Blautia*, *Parabacteroides*, and *Parvimonas micra* were increased, while *Eubacterium eligens* was decreased. *Coprococcus* and *Prevotella* were reduced, and oral species showed a more than two-fold increase in relative abundance, whereas obligate anaerobe genera were reduced. These findings, primarily based on shotgun metagenomics, were largely corroborated by 16S rRNA gene sequencing, except for *Blautia* [37]. Increased exposure to early infections and factors affecting microbiota colonization may increase childhood leukemia risk for both ALL or AML [20,38] (Figure 2).

## 4. Antibiotic Regimens and Outcome in Pediatric Leukemia

A quality improvement project successfully reduced broad-spectrum antibiotic use in children undergoing their first allogeneic HSCT (allo-HSCT) by implementing a short-course empiric antibiotic (EA) regimen based on strict eligibility criteria. The protocol reduced median EA duration from 17 to 8 days and total antibiotic use from 20 to 10 days, with no bloodstream infections, ICU admissions, or deaths, highlighting its safety and potential for broader validation in clinical trials [39]. Research like this has raised the question of whether excessive use of antibiotics could lead to worse outcomes. Experts now recommend adopting short-course empiric antibiotic strategies to mitigate the risks of antimicrobial resistance, infections, and gut microbiome disruption in children with cancer or undergoing allo-HSCT [40]. A report found that antibiotic prophylaxis with trimethoprim-sulfamethoxazole or levofloxacin during induction therapy for ALL increased the short- and medium-term risk of colonization with resistant bacteria to these antibiotics [41]. A meta-analysis of eight RCTs involving 662 febrile neutropenia episodes found no significant differences between short- and long-course antibiotic therapy for all-cause mortality or clinical failure, though short-course therapy reduced fever duration and total antibiotic use by 3–7 days. [42]. However, a balance between infection control and microbiome disruption needs to be maintained since a multicenter, nationwide study revealed significantly higher risks and lower survival rates for bacterial, fungal, and viral infections in pediatric HSCT patients compared to conventional oncological patients, with allo-HSCT patients facing the greatest infection burden and higher incidence of multidrug-resistant bacteria [43]. ALL induces structural changes in the gut microbiota, significantly reducing *alpha* diversity, especially in the presence of antibiotics, but not affecting *beta* diversity. *Bacteroidales* and *Enterococcaceae* were identified as potential biomarkers for ALL, with high predictive value [33].

Implementation of a short-course empiric antibiotic regimen in children undergoing their first allogeneic HSCT (allo-HSCT) proved effective in reducing both the duration of antibiotic therapy and the total amount used without compromising patient safety, as evidenced by the absence of bloodstream infections, ICU admissions, or deaths. This approach aligns with growing concerns about the negative consequences of excessive antibiotic use, including antimicrobial resistance, microbiome disruption, the development of antibiotic-resistant infections, and the risk of aGvHD [44]. Based on data from the Center for International Blood and Marrow Transplant Research and the Pediatric Health Information Services database found that exposure to carbapenems, one class of broad-spectrum antibiotics, was significantly associated with an increased risk of grades 2–4 and grades 3–4 aGVHD. This risk was particularly pronounced when carbapenems were administered before transplantation (prior to day 0) [45]. However, the observed association is likely influenced by clinical decision-making rather than a specific effect of carbapenems. Patients receiving carbapenems were often not previously exposed to other antibiotics, and multi-drug resistance in children is relatively rare. Therefore, the increased risk of aGVHD in these patients may be due to the clinical context, where carbapenems are often reserved for more complex cases or when there is concern for multi-drug resistant infections [45].

## 5. Impact of Chemotherapy on Gut Microbiota and Infection Risks in Pediatric Leukemia

Regarding conventional chemotherapy, some reports reveal that in children with newly diagnosed ALL, a baseline gut microbiome dominated by *Proteobacteria* predicts febrile neutropenia, while *Enterococcaceae* or *Streptococcaceae* dominance during chemotherapy predicts subsequent infections. Chemotherapy significantly alters microbial diversity, decreasing *Bacteroidetes* and increasing taxa like *Clostridiaceae* and *Streptococcaceae*, further influencing infection risks [46]. A study found that altered gut microbiome composition in pediatric oncology patients (including leukemia) was associated with bloodstream infections (BSIs) and differed based on disease type, chemotherapy, and subsequent BSI development. Although no baseline microbiota differences were found for *Clostridium difficile* infection, specific bacterial groups varied significantly between patients with and without post-admission BSI [47]. Moreover, a case–control study found significant differences in the α- and β-diversity of gut microbiota between pediatric ALL patients with and without pneumonia, with *Enterococcus malodoratus*, *Ochrobactrum anthropi*, and *Actinomyces cardiffensis* being significantly more abundant in the affected group. The study also revealed the enrichment of bacterial secretion system pathways in the pneumonia group, highlighting the potential role of gut microbiota alterations in chemotherapy-induced pneumonia in pediatric ALL patients [48]. Specific ratios of facultative and strict anaerobes, such as *Rothia* sp., *Enterocloster*, and *Blautia* spp., were found to predict bacterial infections, bacteremias, and viral enterocolitis after adjusting for demographics and treatment factors. These microbial phylotype ratios were also linked to functional pathways, including lipopolysaccharide production and anaerobic nucleotide biosynthesis [49].

Delayed neutrophil recovery in children with newly diagnosed ALL was associated with decreased abundance of specific gut microbiota genera, including *Ruminococcaceae* and *Lachnospiraceae*, and overgrowth of *Enterococcus*, along with increased chemokine signaling, suggesting that gut dysbiosis may contribute to prolonged neutropenia, highlighting the need for future research on gut-sparing strategies to improve immune recovery [50].

Another research delved deeper into the metabolic pathways of the gut microbiota in children with ALL undergoing chemotherapy, identifying significant alterations in pathways such as pyruvate fermentation to acetate and lactate, and the assimilatory sulfate reduction pathway. These metabolic changes, along with shifts in microbial diversity and composition, were linked to the prediction of infectious complications, with *Bifidobacterium longum* at baseline predicting infections during the first month of chemotherapy [51].

A research paper found that alterations in the gut microbiota, such as reduced *alpha* diversity and increased abundance of unclassified *Enterococcus* and *Lachnospiraceae* species, were linked to systemic inflammation and enterocyte loss during chemotherapy in children with newly diagnosed ALL, therefore predicting the risk of intestinal mucositis [52]. In an animal model of T-ALL, researchers investigated changes in the intestinal and fecal microbiomes and their roles in the development of BSI. The findings showed that BSI in ALL was associated with an increase in mucin-degrading bacteria (*Akkermansia muciniphila*) and a decrease in butyrate-producing *Clostridia* species, along with reduced SCFA levels and altered tight junction protein expression in the small intestine. Functional analysis indicated a diminished ability for SCFA synthesis, and SCFA supplementation helped reduce BSI development in the model [53].

The long-term effects of childhood ALL treatment include alterations in the gut microbiota composition of survivors, characterized by reduced microbial diversity. These changes are associated with immune dysregulation, including increased T cell activation and chronic inflammation, marked by elevated plasma concentrations of IL-6, CRP, and specific T cell (HLA-DR+CD4+ and HLA-DR+CD8+ T cells), and a microbial community enriched for Actinobacteria while depleted of Faecalibacterium [54].

## 6. Gut Microbiome and GvHD in Pediatric Leukemia Patients Post-HSCT

GVHD is a frequent complication following allo-HSCT, triggered by the activation of donor immune cells against the host’s tissues, causing organ damage. Acute GVHD (aGVHD) typically affects the gastrointestinal (GI) tract, skin, and liver, and can involve either a single organ or multiple organs. When the lower GI (LGI) tract is affected, patients have a lower chance of responding to GVHD treatment and face a higher risk of mortality [55,56,57]. Conversely, GVHD restricted to the upper GI (UGI) tract or skin tends to respond well to treatment and does not significantly impact survival outcomes [58,59].

Studies on the impact of microbiome on GVHD have shown that the gut microbiota undergoes significant structural and functional disruption following HSCT, with recovery occurring over the subsequent months. A report showed that the onset of aGvHD is associated with distinct microbiota signatures, with non-aGvHD patients showing higher abundances of propionate-producing *Bacteroidetes* prior to HSCT [60]. Moreover, pediatric HSCT patients who develop GVHD experience a significant decline in gut anti-inflammatory clostridia (AIC) levels, which is associated with cumulative antibiotic exposure, particularly antibiotics effective against anaerobic bacteria. In a preclinical GVHD model, clindamycin depletion of AIC exacerbated GVHD, while oral supplementation of AIC mitigated its severity [61] (Figure 3).

Overall, the development of GvHD has been linked to the enrichment of bacterial families such as *Enterococcaceae*, *Staphylococcaceae*, *Streptococcaceae*, and *Pseudomonadota*. Specific taxa, including *Prevotella* spp. and *Escherichia coli*, have been identified as potential risk factors [62]. Pediatric patients with GvHD often exhibit deficiencies in *Lactobacillus johnsonii*, *Clostridium leptum*, and *Eubacterium rectale*, while adult patients commonly lack Bacteroides and Parabacteroides genera [60,63]. Additionally, a reduced abundance of *Blautia* and *Akkermansia muciniphila*, early after HSCT, has been associated with GvHD onset [64].

Research has shown the emergent role of multidrug-resistant (MDR) beneficial bacteria, such as *Bacteroides fragilis*, *Ruminococcus gnavus*, and *Turicibacter*, in preserving gut microbiome diversity and enhancing immunosurveillance, potentially mitigating GvHD in pediatric ALL patients undergoing HSCT. Conversely, *Enterococcus faecium* strains carrying multiple mutations, were associated with reduced microbial diversity and an increased risk of GvHD [65] (Figure 3).

Some reports go even further, showing that the donor microbiome influences the risk of acute GvHD, with higher bacterial diversity in the donor associated with a reduced risk of acute gastrointestinal GvHD in transplant recipients. Additionally, lower fecal bacterial diversity in recipients was linked to higher mortality rates, suggesting the importance of donor microbiota composition in HSCT outcomes [66]

A very interesting finding was revealed by a systematic review in which gut decontamination (GD) was associated with a lower risk of acute GVHD in allo-HSCT patients pediatric patients, while in adults, the aGvHD was positively correlated with GD. Overall, higher microbiota diversity significantly improved overall survival (OS) and reduced treatment-related mortality (TRM) [67]. Selective gut decontamination (SGD) and total gut decontamination (TGD) during HSCT can impact gut microbiota structure, with SGD maintaining a stable *Bacteroides*-dominated microbiome, while TGD causes greater variability and allows opportunistic pathogens like *Enterococcus* and *Streptococcus* to thrive. These decontamination strategies influence microbial diversity, but their clinical implications on GVHD and overall outcomes are still limited [68]. Microbiome members correlate with biological markers, with high *Lactobacillaceae* abundance linked to severe aGvHD and high mortality, while *Ruminococcaceae* abundance was associated with low aGvHD, rapid NK and B cell reconstitution, and low mortality. Additionally, elevated human beta-defensin 2 (hBD2) and high C-reactive protein levels were observed in patients with *Lactobacillaceae* and *Enterobacteriaceae*, respectively, with antibiotic treatment impacting the microbiome [69] (Figure 3).

Overall, gut microbiota diversity before allogeneic hematopoietic stem cell transplantation (allo-HSCT) correlates with overall survival, with higher diversity associated with better survival outcomes and lower incidence of aGVHD. The higher-diversity group, enriched in beneficial microbes like *Ruminococcaceae* and *Oscillospiraceae*, showed significantly improved survival rates, mostly due to a higher abundance of SCFA–producing taxa, compared to the lower-diversity group, which had an overabundance of *Enterococcaceae* and *Enterobacteriaceae* [70]. Moreover, lower Clostridia abundance, reduced butyrate producers, and lower strict-to-facultative anaerobe ratios, serve as a biomarker for gastrointestinal involvement and survival outcomes in allo-HSCT recipients with acute graft-versus-host disease [71](Figure 3).

## 7. Gut Microbiota Interventions with Pre-, Pro- and Postbiotics in Leukemia Pediatric Patients

With the growing understanding of the gut microbiota ecosystem, numerous studies and publications have highlighted that changes and reductions in gut microbiome diversity may contribute to post-transplant complications, including aGVHD, gastrointestinal issues, and intestinal infections [72,73].

According to the International Scientific Association for Probiotics and Prebiotics (ISAPP), a prebiotic is defined as “a substance that is selectively utilized by host microorganisms to provide a health benefit”[74]. Inulin, found in foods like bananas, onions, and garlic, has been shown to enhance the cytotoxic effects of doxorubicin, potentially reducing the required dose of chemotherapy for the same therapeutic effect [75,76]. Additionally, inulin supplementation improves stool consistency in patients following radiation therapy [77]. Pectin supplementation reduces intestinal damage, improves integrity, and decreases bacterial translocation, while pectic oligosaccharides (POS) alleviate anorexia and reduce adipose tissue loss in leukemia models [78,79]. However, these findings are from animal studies and have not yet been applied to childhood ALL (Table 1).

Currently, probiotic bacteria are available in various products, including foods like yogurt, kefir, and fermented cabbage, as well as dietary supplements and medications. In human nutrition, probiotics are mainly categorized into genera such as *Lactobacillus*, *Bifidobacterium*, *Lactococcus*, *Streptococcus*, and *Enterococcus* [80,81].

Probiotics could mitigate the incidence or severity of GVHD by promoting the generation of regulatory T cells (Tregs), which secrete anti-inflammatory cytokines and modulate the host immune response [82,83]. SCFAs, primarily butyrate, produced by the normal gut microbiota, modulate immune responses by inhibiting the NF-κB signaling pathway, boosting IL-10 expression, and impacting macrophages and dendritic cells [84,85]. By inducing Tregs, these SCFAs may help reduce the severity or occurrence of GVHD [86]. Some reports show modest results regarding the effect of probiotics, like a study where children with ALL recieved *Lactobacillus rhamnosus GG* probiotics significantly decrease in gastrointestinal symptoms and antimicrobial use after chemotherapy was observed. While sepsis and hospitalization rates improved, the differences were not statistically significant [87] (Table 1).

Probiotics have proven effective in treating common gastrointestinal side effects of oncology treatments, such as diarrhea, nausea, vomiting, constipation, and bloating, in many reports. The most commonly used strains include *Lactobacillus acidophilus* and *L. rhamnosus* GG, often in combination with strains like *Bifidobacterium* species, with treatment durations lasting until 6 months [88]. A study in pediatric leukemia patients found that the administration of *Bifidobacterium breve* reduced fever episodes, antibiotic use, and levels of *Enterobacteriaceae* in fecal samples [89]. Another study comparing two antibiotic treatments with and without *Lactobacilli* supplementation found no significant differences in infection rates or recovery times, but the *Lactobacilli* group experienced fewer side effects, such as vomiting and nausea, leading to better medication tolerance [90]. While probiotics can be beneficial for oncology patients, there are reports indicating that oral Lactobacillus treatment may lead to bacteremia in immunocompromised individuals [91,92] (Table 1).

Postbiotics which can optimize physiological functions metabolism and behavior are sometimes referred to as metabiotics. The definition of postbiotics is “a preparation of inanimate microorganisms and/or their components that confers a health benefit on the host” although some researchers also include probiotics’ metabolites and signaling molecules [93,94]. Butyrate plays a dual role in maintaining intestinal barrier function. At low concentrations, it enhances barrier integrity by increasing transepithelial electrical resistance (TER) and reducing permeability. However, at high concentrations, butyrate impairs barrier function by inducing apoptosis and reducing cell viability, leading to increased permeability. This paradoxical effect suggests that while low levels of butyrate are beneficial, excessive butyrate may be toxic and contribute to conditions like neonatal necrotizing enterocolitis (NEC) [95]. Moreover, butyrate mitigates the clinical and pathological effects of Clostridium difficile-induced colitis without affecting bacterial colonization or toxin production. It reduces intestinal inflammation, enhances barrier function, and prevents bacterial translocation by increasing the expression of tight junction proteins. These protective effects are mediated through the stabilization of HIF-1 (hypoxia inducible factor-1) in intestinal epithelial cells, highlighting butyrate’s role in counteracting toxin-induced damage and systemic inflammation [96].

Butyrate, a short-chain fatty acid produced by gut microbes, plays a key role in promoting the differentiation of colonic regulatory T (Treg) cells, which are crucial for suppressing inflammation and allergic responses. An imbalance in gut microbiota, often results in a reduced number of butyrate-producing bacteria, which contributes to the compromised gut barrier function and inflammation. The loss of butyrate-producing microbes is linked to an increased susceptibility to infections and inflammatory diseases [96].

SCFAs, including acetic acid, butyric acid, and propionic acid, are produced by fermenting polysaccharides in the human intestine. Among them, butyrate has demonstrated anti-cancer activity by inhibiting histone deacetylase, a strategy currently being explored in clinical trials for potential therapeutic use [97,98]. Butyrate (BA) at high concentrations induced apoptosis in human acute leukemia cells, activating caspase-3 and reducing cell viability in U937 cells. It also significantly decreased the chemokines CCL2 and CCL5 in various leukemia cell lines, suggesting BA as a potential therapy to promote apoptosis and regulate cytokine production in cancer [99]. A study examined the impact of enteral nutrition (EN) versus parenteral nutrition (PN) on SCFA production in pediatric patients undergoing HSCT. The findings revealed that EN promoted a quicker recovery of SCFA production, and supported the restoration of a healthy gut microbiome and potentially reducing the risk of infections and GvHD [100]. These findings are supported by the results of a meta-analysis, which show that EN reduces the incidence of aGVHD, particularly grade III-IV and gut aGVHD, potentially due to improved gut eubiosis in enterally fed patients [101] (Table 1).

**Table 1 children-12-00166-t001:** Benefits of prebiotics, probiotics, and postbiotic supplementation in childhood leukemia.

Category	Intervention	Description and Benefits	Reference
Prebiotics	Inulin	Enhances the cytotoxic effects of doxorubicin, potentially reducing chemotherapy dose; improves stool consistency following radiation therapy.	[70,71,72,73]
	Pectin supplementation	Reduces intestinal damage, improves integrity, decreases bacterial translocation, and alleviates anorexia in leukemia models.	[74,75]
Probiotics	*Lactobacillus rhamnosus* GG	Reduces gastrointestinal symptoms and antimicrobial use post-chemotherapy; shows modest improvements in sepsis and hospitalization rates.	[83]
	*Bifidobacterium breve*	Decreases fever episodes, antibiotic use, and Enterobacteriaceae levels in fecal samples of pediatric leukemia patients.	[85]
	*Lactobacillus acidophilus* and *L. rhamnosus* GG	Effective in treating diarrhea, nausea, vomiting, constipation, and bloating associated with oncology treatments. Often used with Bifidobacterium species for up to 6 months.	[84]
Postbiotics	Butyrate (BA)	Induces apoptosis in leukemia cells by activating caspase-3; reduces chemokines (CCL2, CCL5), promoting apoptosis and cytokine regulation.	[91,92,93]
	SCFAs (Short-Chain Fatty Acids)	Modulates immune responses, inhibits NF-κB signaling, boosts IL-10 expression, and impacts macrophages and dendritic cells. SCFAs may reduce GVHD severity by inducing regulatory T cells.	[80,81,82,94]
Nutrition Approach	Enteral Nutrition (EN)	Supports quicker SCFA production recovery, restores gut microbiome health, and reduces infection and GVHD risk compared to parenteral nutrition (PN).	[94,95]

## 8. Fecal Microbiota Transplantation Following Childhood Leukemia Treatment

FMT, despite concerns about its safety and acceptability, has been used to treat microbiome-related conditions like Clostridium difficile infections (CDIs), irritable bowel syndrome (IBS), and inflammatory bowel disease (IBD). Moreover, FMT has been used to treat many other gut-unrelated conditions [102]. Current evidence shows little to no data on the role of FMT in alleviating complications of pediatric ALL, but one report highlights its effects in gut GVHD. By day 120, six of seven patients achieved clinical responses, with improved gut microbiota, including *Bacteroides fragilis* and *Faecalibacterium prausnitzii*, and stable immune cell levels in survivors. Adverse events, such as nausea and abdominal pain, occurred in 86% but were mostly mild, while bacterial mass recovery suggests potential microbiota restoration [103]. Promising results come from a study of adults and children with GI GVHD, where FMT led to significantly higher bacterial mass and increased levels of *Bifidobacterium* spp., Escherichia coli, and Bacteroides fragilis compared to the placebo group. FMT also resulted in faster partial response (4 days vs. 48 days) and higher complete response rates at 30, 60, and 90 days (42%, 74%, and 84%, respectively), with adverse events being similar between both groups [104]. Moreover, in a case report, a pediatric patient with T-cell lymphoblastic leukemia and acute gastrointestinal GvHD underwent multiple FMT procedures after failing previous treatments and suffering from multi-resistant bacterial colonization. The FMT, administered via gastroscopy under general anesthesia, resulted in clinical improvement following the treatments [105] (Table 2).

Other reports have evaluated the benefit of FMT in MDR bacteria colonization among pediatric ALL patients, showing an 80% decolonization rate within one week, though recurrent colonization occurred in four of five patients by the last follow-up. Adverse events were mild and transient, except for one sepsis episode attributed to the same colonizing pathogen treated by FMT, resolved with antibiotics. These findings suggest FMT’s feasibility and safety, though repeated procedures may enhance long-term decolonization in high-risk patients [106]. Another case report demonstrated successful FMT in immunocompromised children with refractory diarrhea. Pre-FMT stool samples revealed *Proteobacteria* as the dominant phylum, which decreased gradually after the procedure, while Firmicutes increased over time. Post-FMT, the microbial community showed improved diversity, as evidenced by a rapid rise in Shannon’s diversity index [107]. In a case report of three pediatric HSCT recipients with recurrent CDI, FMT was safe and well-tolerated, with only mild adverse effects like nausea and vomiting. While one patient with AML achieved successful CDI clearance, recurrence occurred in two others, suggesting that factors like chemotherapy and antimicrobial use may limit FMT efficacy in this population [108] (Table 2).

In a mixed study of 80 patients treated with FMT for recurrent, refractory, or severe CDI, the overall cure rate was 89%, with 78% achieving resolution after a single FMT. Among pediatric patients, the outcomes were consistent with the overall group, demonstrating efficacy and no infections definitively linked to FMT, though close monitoring is crucial for immunocompromised children due to their higher risk of complications [109]. Another report analyzed immunocompromised pediatric patients with both malignant (including hematological) and non-malignant conditions who underwent FMT for recurrent CDI, showing a 79% success rate after the first FMT and 86% after one or more treatments. Although 31% of patients experienced serious adverse events, none were fatal or associated with multi-drug resistant infections, and all patients fully recovered [110] (Table 2).

The safety of FMT in children was evaluated in a study, with short-term adverse events (26.32%) being mostly mild and self-limiting, including abdominal pain, diarrhea, and fever. Severe AEs were rare, though immunodeficiency, a common concern in pediatric ALL patients, significantly increased the risk, suggesting the need for cautious patient selection and monitoring when considering FMT in this vulnerable group [111] (Table 2).

## 9. Conclusions

The gut microbiome plays a crucial role in the outcomes of pediatric leukemia patients, particularly those undergoing HSCT. Dysbiosis, caused by chemotherapy, antibiotics, and immune changes, is linked to complications like GVHD, gastrointestinal issues, and infections. Interventions such as prebiotics, probiotics, postbiotics, and FMT show promise in restoring microbial balance, reducing GVHD severity, and improving survival. While initial results are encouraging, further research is needed to establish safe and effective protocols. Careful patient selection and monitoring are essential, particularly in immunocompromised children, to maximize the benefits of microbiome-targeted therapies in leukemia treatment.

## Figures and Tables

**Figure 1 children-12-00166-f001:**
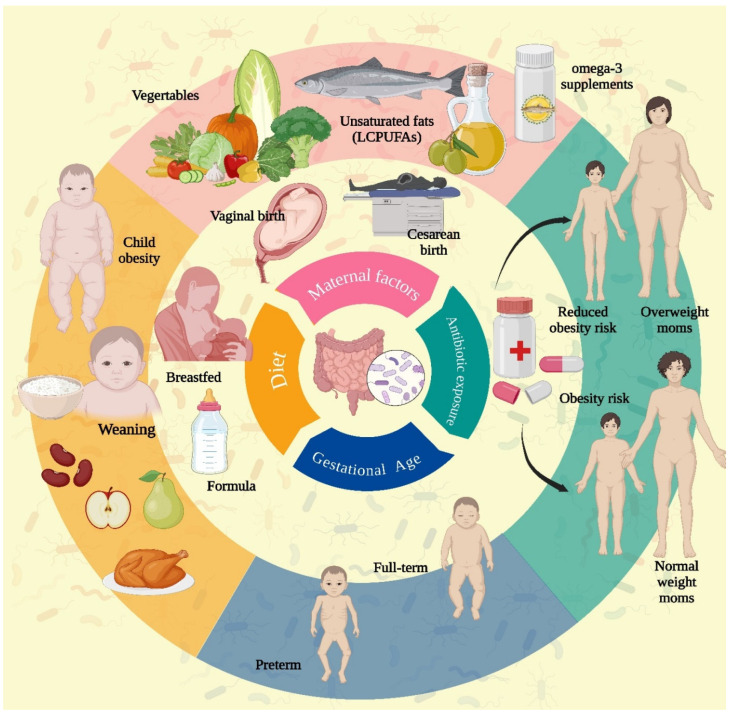
Factors influencing dynamics of child gut-microbiome.

**Figure 2 children-12-00166-f002:**
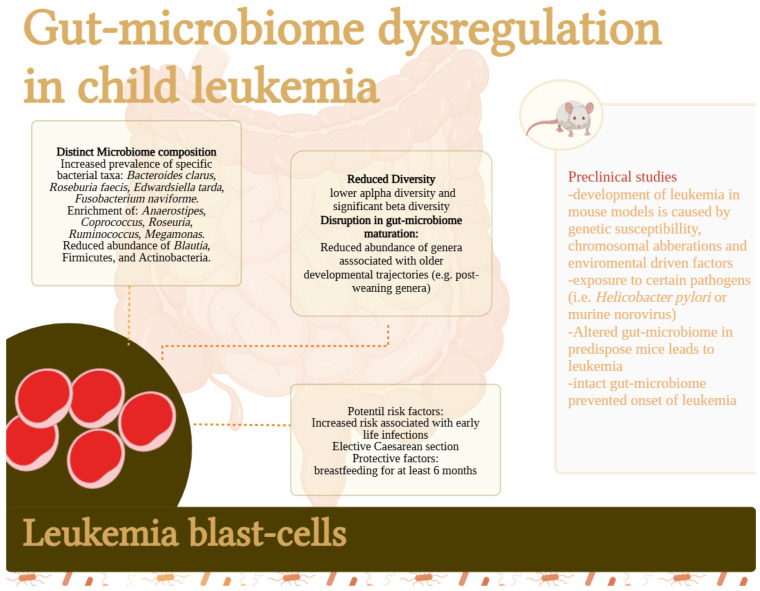
Gut microbiome and leukemia risk in children.

**Figure 3 children-12-00166-f003:**
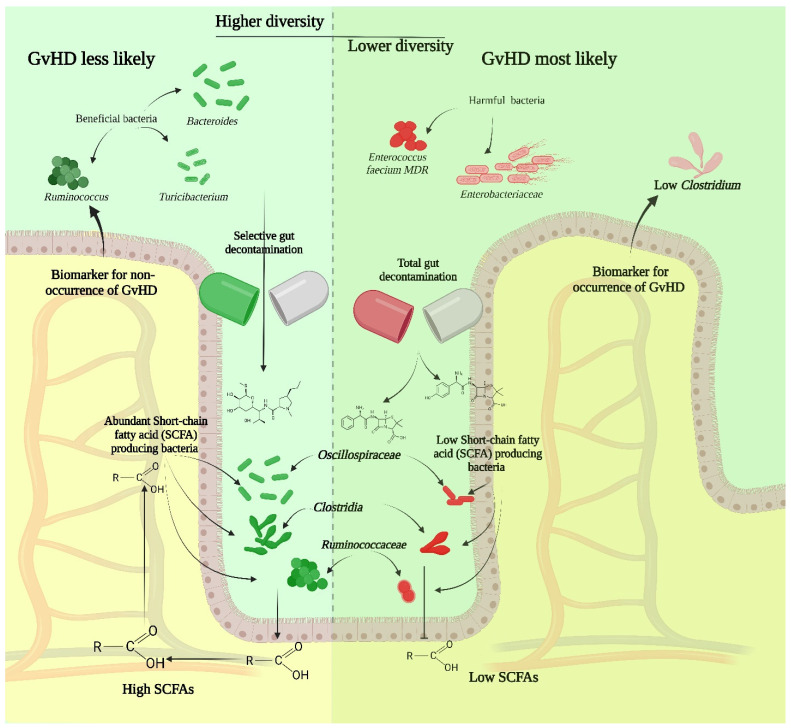
GvHD occurrence and gut microbiome changes.

**Table 2 children-12-00166-t002:** Fecal microbiota transplant in children with leukemia.

Study	Patient Type	Intervention Details	Outcomes	Adverse Events	Comments
Pavlov University, St. Petersburg, Russia; et al. [103]	Pediatric patients with gut GVHD	FMT administered; improved gut microbiota (*Bacteroides fragilis*, *Faecalibacterium prausnitzii*)	6 of 7 patients had clinical responses and stable immune cells	Nausea, abdominal pain (86%)—mostly mild	Suggests microbiota restoration with FMT
Goloshchapov et al. [104]	Adults and children with GI GVHD	FMT administered; increased bacterial mass, *Bifidobacterium* spp., *Escherichia coli*, *Bacteroides fragilis*	Faster response (4 days vs. 48 days), improved complete response rates (42%, 74%, 84% at 30, 60, 90 days)	Similar adverse events in both groups	Promising for GI GVHD treatment
Fałkowska et al. [105] (Case report)	Pediatric patient with T-cell lymphoblastic leukemia and GI GVHD	Multiple FMT procedures after failed treatments and multi-resistant bacterial colonization	Clinical improvement after FMT	No significant adverse events	Highlights FMT’s potential in refractory cases
Merli et al. [106]	Pediatric ALL patients with MDR bacteria colonization	FMT administered for decolonization	80% decolonization rate within 1-week, recurrent colonization in 4/5 patients	Mild, transient adverse events; one sepsis episode treated with antibiotics	FMT shows feasibility, repeated procedures may help with long-term decolonization
Zhong et al. [107]	Immunodeficient children with refractory diarrhea	FMT administered	Improved microbial diversity post-FMT (Shannon’s diversity index)	No significant adverse events	Demonstrates FMT’s potential for gut health restoration
Bluestone et al. [108]	Pediatric HSCT recipients with recurrent CDI	FMT administered	Safe and well-tolerated; CDI clearance in 1 patient, recurrence in 2 others	Mild adverse effects (nausea, vomiting)	Efficacy may be limited by chemotherapy and antimicrobial use
Kelly et al. [109]	80 patients (75 adults, 5 children) with recurrent, refractory, or severe CDI	FMT administered	89% cure rate; 78% resolution after a single FMT	No infections definitively linked to FMT, close monitoring needed	Positive results in both pediatric and adult populations
Conover et al. [110]	Immunocompromised pediatric patients with malignant and non-malignant conditions	FMT for recurrent CDI	79% success after 1st FMT, 86% after multiple treatments	31% serious adverse events; no fatal events	High success rate but caution needed for immunocompromised children
Zhang et al. [111]	Pediatric patients	FMT administered	Short-term adverse events (26.32% mild and self-limiting)	Abdominal pain, diarrhea, fever	Immunodeficiency increases risk, needs careful monitoring

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
