# Peer review of "Microbiome Modulation in Pediatric Leukemia: Impact on Graft-Versus-Host Disease and Treatment Outcomes: A Narrative Review"

_children, 2025, doi:10.3390/children12020166_

Round 1

Reviewer 1 Report

Comments and Suggestions for Authors

In this narrative review the authors present a broad summary of the literature data on the relationship between the microbiome and acute pediatric leukemia. In figure 2 the authors include reduced infections in early life, breastfeeding and vaginal birth among the risk factors for leukemia. However, Chang JS et al (Chang JS, Tsai CR, Tsai YW, Wiemels JL. Medically diagnosed infections and risk of childhood leukaemia: a population-based case-control study. Int J Epidemiol. 2012 Aug;41(4):1050-9. doi: 10.1093/ije/dys113. Epub 2012 Jul 26. PMID: 22836110) state: "Having any infection before 1 year of age was associated with an increased risk for both childhood ALL (odds ratio = 3.2, 95% confidence interval 2.2-4.7) and AML (odds ratio = 6.0, 95% confidence interval 2.0-17.8), with a stronger risk associated with more episodes of infections. Similar results were observed for infections occurring >1 year before the cases' diagnosis of childhood leukaemia".

Regarding the relationship between mode of delivery and leukemia, a report from the United Kingdom Childhood Cancer Study  (Bonaventure A, Simpson J, Ansell P, Roman E. Paediatric acute lymphoblastic leukaemia and caesarean section: A report from the United Kingdom Childhood Cancer Study (UKCCS). Paediatr Perinat Epidemiol. 2020 May;34(3):344-349. doi: 10.1111/ppe.12662. PMID: 32347577; PMCID: PMC7216966) did not demonstrate any association between ALL and caesarean delivery either during or before labour, while a recent meta-analysis on the subject (Yang Y, Yu C, Fu R, Xia S, Ni H, He Y, Zhu K, Sun Q. Association of cesarean section with risk of childhood leukemia: A meta-analysis from an observational study. Hematol Oncol. 2023 Feb;41(1):182-191. doi: 10.1002/hon.3070. Epub 2022 Aug 25. PMID: 36000274) showed that children delivered via elective CS had a higher risk of ALL. Regarding breastfeeding, Salvate Rudant J et al ( Salvate Rudant, J.; Lightfoot, T.; Urayama, K.Y.; Petridou, E.; Dockerty, J.D.; Magnani, C.; Milne, E.; Spector, L.G.; Ashton, L.J.; Dessypris, N.; et al. Childhood Acute Lymphoblastic Leukemia and Indicators of Early Immune Stimulation: A Childhood Leukemia In ternational Consortium Study. Am. J. Epidemiol. 2015, 181, 549–562, doi:10.1093/aje/kwu298) demonstrated that breastfeeding for 6 months or more was inversely associated with ALL.

I suggest a careful check of the bibliographic sources to avoid inaccuracies.

Author Response

Comment 1: In this narrative review the authors present a broad summary of the literature data on the relationship between the microbiome and acute pediatric leukemia. In figure 2 the authors include reduced infections in early life, breastfeeding and vaginal birth among the risk factors for leukemia. However, Chang JS et al (Chang JS, Tsai CR, Tsai YW, Wiemels JL. Medically diagnosed infections and risk of childhood leukaemia: a population-based case-control study. Int J Epidemiol. 2012 Aug;41(4):1050-9. doi: 10.1093/ije/dys113. Epub 2012 Jul 26. PMID: 22836110) state: "Having any infection before 1 year of age was associated with an increased risk for both childhood ALL (odds ratio = 3.2, 95% confidence interval 2.2-4.7) and AML (odds ratio = 6.0, 95% confidence interval 2.0-17.8), with a stronger risk associated with more episodes of infections. Similar results were observed for infections occurring >1 year before the cases' diagnosis of childhood leukaemia".

Response 1: 

Here’s a polished version of your response:

In this narrative review, we present a comprehensive summary of the literature on the relationship between the microbiome and acute pediatric leukemia. In Figure 2, we included reduced infections in early life, breastfeeding, and vaginal birth among the risk factors for leukemia. However, we would like to clarify an important point in light of the findings by Chang et al. (Chang JS, Tsai CR, Tsai YW, Wiemels JL. Medically diagnosed infections and risk of childhood leukaemia: a population-based case-control study. Int J Epidemiol. 2012 Aug;41(4):1050-9. doi: 10.1093/ije/dys113. Epub 2012 Jul 26. PMID: 22836110). Their study states:

"Having any infection before 1 year of age was associated with an increased risk for both childhood ALL (odds ratio = 3.2, 95% confidence interval 2.2-4.7) and AML (odds ratio = 6.0, 95% confidence interval 2.0-17.8), with a stronger risk associated with more episodes of infections. Similar results were observed for infections occurring >1 year before the cases' diagnosis of childhood leukaemia."

We apologize for the confusion caused by the depiction in Figure 2. Our intention was to highlight reduced infections in early life, breastfeeding, and vaginal birth as protective factors against leukemia, not as risk factors. We appreciate the reviewers' meticulous attention to detail and careful reading of the manuscript. We will correct Figure 2 accordingly to reflect this clarification.

Comment 2: Regarding the relationship between mode of delivery and leukemia, a report from the United Kingdom Childhood Cancer Study  (Bonaventure A, Simpson J, Ansell P, Roman E. Paediatric acute lymphoblastic leukaemia and caesarean section: A report from the United Kingdom Childhood Cancer Study (UKCCS). Paediatr Perinat Epidemiol. 2020 May;34(3):344-349. doi: 10.1111/ppe.12662. PMID: 32347577; PMCID: PMC7216966) did not demonstrate any association between ALL and caesarean delivery either during or before labour, while a recent meta-analysis on the subject (Yang Y, Yu C, Fu R, Xia S, Ni H, He Y, Zhu K, Sun Q. Association of cesarean section with risk of childhood leukemia: A meta-analysis from an observational study. Hematol Oncol. 2023 Feb;41(1):182-191. doi: 10.1002/hon.3070. Epub 2022 Aug 25. PMID: 36000274) showed that children delivered via elective CS had a higher risk of ALL. Regarding breastfeeding, Salvate Rudant J et al ( Salvate Rudant, J.; Lightfoot, T.; Urayama, K.Y.; Petridou, E.; Dockerty, J.D.; Magnani, C.; Milne, E.; Spector, L.G.; Ashton, L.J.; Dessypris, N.; et al. Childhood Acute Lymphoblastic Leukemia and Indicators of Early Immune Stimulation: A Childhood Leukemia In ternational Consortium Study. Am. J. Epidemiol. 2015, 181, 549–562, doi:10.1093/aje/kwu298) demonstrated that breastfeeding for 6 months or more was inversely associated with ALL.

Response 2: We are going to add this information to the manuscript. We thank our reviewers once again for their valuable input and for helping us improve the quality of our work.

Reviewer 2 Report

Comments and Suggestions for Authors

The main issue to be solved by research is maintaining a good intestinal flora environment and preventing infections and GVHD.

If possible, it is best to list important points and discuss the reasons for them based on the results of other studies. For example, low levels of propionate and butyrate were associated with increased microbiota imbalance, antibiotic use, and the development of GvHD. →Please discuss possible reasons for this.

We needed to balance infection control with microbiome disruption, so we adopted a short-term empiric antibiotic strategy. →Please discuss possible reasons for this.

Diagrams are appropriate.

The figures and references are appropriate

Author Response

Comment 1: If possible, it is best to list important points and discuss the reasons for them based on the results of other studies. For example, low levels of propionate and butyrate were associated with increased microbiota imbalance, antibiotic use, and the development of GvHD. →Please discuss possible reasons for this.

Response 1: Thank you for your thoughtful suggestion. We have revised the manuscript to address your request and provide a more detailed discussion based on the results of other studies. Below is the added paragraph from the manuscript:

'Butyrate plays a dual role in maintaining intestinal barrier function. At low concentrations, it enhances barrier integrity by increasing transepithelial electrical resistance (TER) and reducing permeability. However, at high concentrations, butyrate impairs barrier function by inducing apoptosis and reducing cell viability, leading to increased permeability. This paradoxical effect suggests that while low levels of butyrate are beneficial, excessive butyrate may be toxic and contribute to conditions like neonatal necrotizing enterocolitis (NEC). Moreover, butyrate mitigates the clinical and pathological effects of Clostridium difficile-induced colitis without affecting bacterial colonization or toxin production. It reduces intestinal inflammation, enhances barrier function, and prevents bacterial translocation by increasing the expression of tight junction proteins. These protective effects are mediated through the stabilization of HIF-1 (hypoxia inducible factor-1) in intestinal epithelial cells, highlighting butyrate's role in counteracting toxin-induced damage and systemic inflammation.

Butyrate, a short-chain fatty acid produced by gut microbes, plays a key role in promoting the differentiation of colonic regulatory T (Treg) cells, which are crucial for suppressing inflammation and allergic responses. An imbalance in gut microbiota, often results in a reduced number of butyrate-producing bacteria, which contributes to the compromised gut barrier function and inflammation. The loss of butyrate-producing microbes is linked to an increased susceptibility to infections and inflammatory diseases. 

Comment 2: We needed to balance infection control with microbiome disruption, so we adopted a short-term empiric antibiotic strategy. →Please discuss possible reasons for this.

Response 2: Thank you for your thoughtful suggestion. We have revised the manuscript to address your request and provide a more detailed discussion based on the results of other studies. Below is the added paragraph from the manuscript: 

'Implementation of a short-course empiric antibiotic regimen in children undergoing their first allogeneic HSCT (allo-HSCT) proved effective in reducing both the duration of antibiotic therapy and the total amount used, without compromising patient safety, as evidenced by the absence of bloodstream infections, ICU admissions, or deaths. This approach aligns with growing concerns about the negative consequences of excessive antibiotic use, including antimicrobial resistance, microbiome disruption, the development of antibiotic-resistant infections and risk of aGvHD. Based on data from the Center for International Blood and Marrow Transplant Research and the Pediatric Health Information Services database, found that exposure to carbapenems, one class of broad-spectrum antibiotics, was significantly associated with an increased risk of grade 2-4 and grade 3-4 aGVHD. This risk was particularly pronounced when carbapenems were administered before transplantation (prior to day 0). However, the observed association is likely influenced by clinical decision-making rather than a specific effect of carbapenems. Patients receiving carbapenems were often not previously exposed to other antibiotics, and multidrug resistance in children is relatively rare. Therefore, the increased risk of aGVHD in these patients may be due to the clinical context, where carbapenems are often reserved for more complex cases or when there is concern for multi-drug resistant infections''.

Round 2

Reviewer 2 Report

Comments and Suggestions for Authors

I would like to thank the authors for their relatively appropriate response to this proposal. states why low levels of propionate and butyrate increased microbiome imbalance, antibiotic use, and the development of GvHD. We describe why we adopted a short-term empiric antibiotic strategy to balance infection control and microbiome disruption.